# On the Predictability of Pruning Across Scales

## Abstract

We show that the error of iteratively-pruned networks empirically follows a scaling law with interpretable coefficients that depend on the architecture and task. We functionally approximate the error of the pruned networks, showing that it is predictable in terms of an invariant tying width, depth, and pruning level, such that networks of vastly different sparsities are freely interchangeable. We demonstrate the accuracy of this functional approximation over scales spanning orders of magnitude in depth, width, dataset size, and sparsity. We show that the scaling law functional form holds (generalizes) for large scale data (CIFAR-10, ImageNet), architectures (ResNets, VGGs) and iterative pruning algorithms (IMP, SynFlow). As neural networks become ever larger and more expensive to train, our findings suggest a framework for reasoning conceptually and analytically about pruning.

## 1 Introduction

For decades, neural network *pruning*—eliminating unwanted parts of the network—has been a popular approach for reducing network sizes or computational demands of inference (LeCun et al., 1990; Reed, 1993; Han et al., 2015). In practice, pruning can reduce the parameter-counts of contemporary models by 2x (Gordon et al., 2020) to 5x (Renda et al., 2020) with no reduction in accuracy. More than 80 pruning techniques have been published in the past decade (Blalock et al., 2020), but, despite this enormous volume of research, there remains little guidance on important aspects of pruning. Consider a seemingly simple question one might ask when using a particular pruning technique:

*Given a family of neural networks (e.g., ResNets on ImageNet of various widths and depths), which family member should we prune (and by how much) to obtain the network with the smallest parameter-count such that error does not exceed some threshold $\epsilon_k$?*

As a first try, we could attempt to answer this question using brute force: we could prune every member of a family (i.e., perform grid search over widths and depths) and select the smallest pruned network that satisfies our constraint on error. However, depending on the technique, pruning one network (let alone grid searching) could take days or weeks on expensive hardware.

If we want a more efficient alternative, we will need to make assumptions about pruned networks: namely, that there is some *structure* to the way that their error behaves. For example, that pruning a particular network changes the error in a predictable way. Or that changing the width or depth of a network changes the error when pruning it in a predictable way. We could then train a smaller number of networks, characterize this structure, and estimate the answer to our question.

We have reason to believe that such structure does exist for pruning: techniques already take advantage of it implicitly. For example, Cai et al. (2019) create a single neural network architecture that can be scaled down to many different sizes; to choose which subnetwork to deploy, Cai et al. train an auxiliary, black-box neural network to predict subnetwork performance. Although this black-box approach implies the existence of structure, it does not reveal this structure explicitly or make it possible to reason analytically in a fashion that could answer our research question.

Outside the context of pruning algorithms, such structure has been observed—and further codified explicitly—yielding insights and predictions in the form of scaling laws. Tan and Le (2019) design the *EfficientNet* family by developing a heuristic for predicting efficient tradeoffs between depth, width, and resolution. Hestness et al. (2017) observe a power-law relationship between dataset size

and the error of vision and NLP models. Rosenfeld et al. (2020) use a power scaling law to predict the error of all variations of architecture families and dataset sizes jointly, for computer vision and natural language processing settings. Kaplan et al. (2020) develop a similar power law for language models that incorporates the computational cost of training.

Inspired by this line of work, we address our research question about pruning by developing a scaling law to predict the error of networks as they are pruned. To the best of our knowledge, no explicit scaling law holding over pruning algorithms and network types currently exists. In order to formulate such a predictive scaling law, we consider the dependence of generalization error on the pruning-induced *density* for networks of different depths and width trained on different dataset sizes.

We begin by developing a functional form that accurately estimates the generalization error of a specific model as it is pruned (Section 3). We then account for other architectural degrees of freedom, expanding the functional form for pruning into a scaling law that jointly considers density alongside width, depth, and dataset size (Section 4). The basis for this joint scaling law is an *invariant* we uncover that describes ways that we can interchange depth, width, and pruning without affecting error. The result is a scaling law that accurately predicts the performance of pruned networks across scales. Finally, we use this scaling law to answer our motivating question (Section 7).

The same functional form can accurately estimate the error for both unstructured magnitude pruning (Renda et al., 2020) and SynFlow (Tanaka et al., 2020) when fit to the corresponding data, suggesting we have uncovered structure that may be applicable to iterative pruning more generally. And now that we have established this functional form, fitting it requires only a small amount of data (Appendix 5). In summary, our contributions are as follows:

- We develop a scaling law that accurately estimates the error when pruning a single network.
- We observe and characterize an *invariant* that allows error-preserving interchangeability among depth, width, and pruning density.
- Using this invariant, we extend our single-network scaling law into a joint scaling law that predicts the error of all members of a network family at all dataset sizes and all pruning densities.
- In doing so, we demonstrate that there is structure to the behavior of the error of iteratively pruned networks that we can capture explicitly with a simple functional form.
- Our scaling law enables a framework for reasoning analytically about pruning, allowing us to answer our motivating question and similar questions about pruning.

## 2 EXPERIMENTAL SETUP

**Pruning.** We study two techniques for pruning neural networks: *iterative magnitude pruning* (IMP) (Janowsky, 1989; Han et al., 2015; Frankle et al., 2020) in the main body of the paper and *SynFlow* (Tanaka et al., 2020) in Appendix E. We describe IMP in detail here and SynFlow in Appendix A. IMP prunes by removing a fraction—typically 20%, as we do here—of individual weights with the lowest magnitudes at the end of training.[1] We choose these weights globally throughout the network, i.e., without regard to specific layers. We use per-weight magnitude pruning because it is generic, well-studied (Han et al., 2015), and matches the sparsity/accuracy tradeoffs of more complicated methods (Gale et al., 2019; Blalock et al., 2020; Renda et al., 2020).

Pruning weights typically reduces the accuracy of the trained network, so it is standard practice to further train after pruning to recover accuracy. For IMP, we use a practice called *weight rewinding*, in which the values of unpruned weights are *rewound* to their values at epoch 10 and the training process is repeated from there to completion. To achieve density levels below 80%, this process is repeated *iteratively*—pruning by 20%, rewinding, and retraining—until a desired density level is reached. Renda et al. (2020) demonstrate that IMP with weight rewinding achieves state-of-the-art tradeoffs between sparsity and accuracy. For a formal statement of this pruning algorithm, see Appendix A.

**Datasets.** We study the image classification tasks CIFAR-10 and ImageNet. Our scaling law predicts the error when training with the entire dataset and smaller *subsamples*. To subsample a dataset to a size of $n$, we randomly select $n$ of the training examples without regard to individual classes such

---

[1]We do not prune biases or BatchNorm, so pruning 20% of weights prunes fewer than 20% of parameters.

that in expectation we preserve the original dataset distribution (we always retain the entire test set). When performing iterative pruning, we maintain the same subsample for all pruning iterations.

**Networks.** We study three families of neural networks: ResNets for CIFAR-10, ResNets for ImageNet, and (in Appendix E) VGG-style networks for CIFAR-10.[2] We develop a scaling law that predicts the error (when pruned) of an entire *family* of networks with varying widths and—in the case of the CIFAR-10 ResNets—depths. To vary width, we multiply the number of channels in each layer by a *width scaling factor*. To vary depth of the CIFAR-10 ResNets, we vary the number of residual blocks. We refer to a network by its depth $l$ (the number of layers in the network, not counting skip connections) and its width scaling factor $w$.

**Notation and terminology.** Throughout the paper, we use the following notation and terminology:

- $\mathbb{D}_N = \{\boldsymbol{x}_i, y_i\}_{i=1}^N$ is a labeled training set with $N$ examples. A *subsample* of size $n$ is a subset of $\mathbb{D}_N$ containing $n$ examples selected uniformly at random.
- $l$ and $w$ are, respectively, the depth (i.e., the number of layers, excluding skip connections) and the width scaling factor of a particular network.
- A collection of networks that vary by width and depth are a *network family*.
- $s$ is the *sparsity* of a pruned network (i.e., the fraction of weights that have been pruned) and $d \triangleq 1 - s$ is the *density* (i.e., the fraction of weights that have not been pruned).
- $\epsilon(d, l, w, n)$ is the test error of a network with the specified density, depth, width scaling factor, and dataset size.
- $\epsilon_{np}(l, w, n) = \epsilon(1, l, w, n)$ is the test error of the unpruned network with the specified depth, width scaling factor, and dataset size. When clear from context, we omit $(w, l, n)$ and write $\epsilon_{np}$.
- $\hat{\epsilon}(\epsilon_{np}, d \mid l, w, n)$ is an estimate of the error of a pruned model for a scaling law that has been fit to a specific network with the specified depth, width scaling factor, and dataset size (Section 3).
- $\hat{\epsilon}(\epsilon_{np}, d, l, w, n)$ is an estimate of the error of a pruned model with the specified depth, width scaling factor, and dataset size for a scaling law that has been fit to a network family (Section 4).

**Dimensions.** In developing scaling laws, we vary four different dimensions: dataset subsample size ($n$) and network degrees of freedom density ($d$), network depth ($l$), and width scaling factor ($w$). We consider the following ranges of these values in our experiments in the main body of the paper:

| Network Family | $N_{\text{train}}$ | $N_{\text{test}}$ | Densities ($d$) | Depths ($l$) | Width Scalings ($w$) | Subsample Sizes ($n$) |
|---|---|---|---|---|---|---|
| CIFAR-10 ResNet | 50K | 10K | $0.8^i, i \in \{0, \dots, 40\}$ | 8, 14, 20, 26, 50, 98 | $2^i, i \in \{-4, \dots, 2\}$ | $\frac{N}{i}, i \in \{1, 2, 4, 8, 16, 32, 64\}$ |
| ImageNet ResNet | 1.28M | 50K | $0.8^i, i \in \{0, \dots, 30\}$ | 50 | $2^i, i \in \{-4, \dots, 0\}$ | $\frac{N}{i}, i \in \{1, 2, 4\}$ |

We use sanity checks to filter infeasible or unusual configurations from this table. In many cases, networks become disconnected before we reach the lowest density (e.g., ∼30% of the CIFAR-10 ResNet configurations). We also eliminate configurations where increasing width or depth of the unpruned network lowers test accuracy (e.g., 144 of the 294 CIFAR-10 ResNet configurations of $l$, $w$, and $n$); these are typically unusual, imbalanced configurations (e.g., $l = 98$, $w = \frac{1}{16}$). Of the 12,054 possible CIFAR-10 ResNet configurations, about 8,000 are eliminated based on these sanity checks.

## 3 MODELING THE ERROR OF A PRUNED NETWORK

Our goal in this section is to develop a functional form that accurately models the error of a member of a network family as it is pruned (using IMP here and SynFlow in Appendix E) based on its unpruned error $\epsilon_{np}(w, l, n)$. In other words, we wish to find a function $\hat{\epsilon}(\epsilon_{np}, d \mid l, w, n)$ that predicts the error at each density $d$ for a network with a particular depth $l$, width scaling factor $w$, and dataset size $n$.

**Intuition.** Since IMP prunes a network 20% at a time, it produces pruned networks at intermediate levels of density $d_k = 0.8^k$ in the process of creating a final pruned network at density $d_K = 0.8^K$. In Figure 1 (left), we plot the error of these pruned networks for members of the CIFAR-10 ResNet family with a different widths $w$. All of these curves follow a similar pattern:[3]

---

[2]See Appendix B for full details on architectures and hyperparameters.

[3]The same patterns occur when varying $l$ and $n$ for CIFAR-10 and $w$ and $n$ for ImageNet (Appendix C). We focus on varying width for CIFAR-10 here for illustrative purposes.

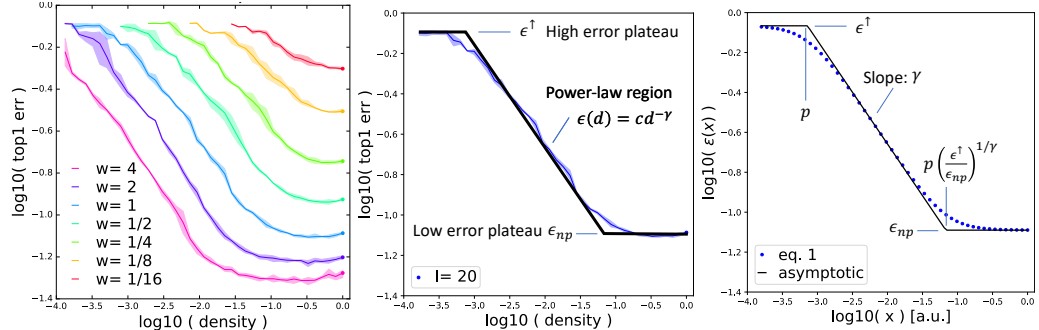

Figure 1: Relationship between density and error when pruning CIFAR-10 ResNets; $w$ varies, $l = 20$, $n = N$ (left). Low-error plateau, power-law region, and high-error plateau for CIFAR-10 ResNet $l = 20$, $w = 1$, $n = N$ (center). Visualizing Equation 1 and the roles of free parameters (right).

*Observation 1: Low-error plateau.* The densest pruned networks (right part of the curves) have approximately the same error as the unpruned network: $\epsilon_{np}(w)$. We call this the *low-error plateau.*

*Observation 2: Power-law region.* When pruned further, error increases in a linear fashion on the logarithmic axes of the figure. Linear behavior on a logarithmic scale is the functional form of a *power law*, in which error relates to density through an exponent $\gamma$ and a coefficient $c$: $\epsilon(d, w) \approx cd^{-\gamma}$. In particular, $\gamma$ controls the slope of the line on the logarithmic axes.

*Observation 3: High-error plateau.* When pruned further, error again flattens; we call this the *high-error plateau* and call the error of the plateau $\epsilon^{\uparrow}$.

Figure 1 (center) labels these regions for CIFAR-10 ResNet-20 (width scaling factor 1, dataset size $N$) and shows an approximation of these regions that is piece-wise linear on logarithmic axes. These observations are our starting point for developing a functional form that estimates error when pruning.

**Functional form.** Our next task is to find a functional form that accurately captures these observations about the relationship between density and error. In prior work, Rosenfeld et al. (2020) observe that the relationship between width and error shares the same general shape: it has a region of lower error, a power-law region, and region of higher error. However, this relationship is different enough from the one we observe (see Appendix G) to merit an entirely new functional form.

To develop this functional form, we note that the three regions of the curves in Figure 1 (the low-error plateau, the power-law region, and the high-error plateau) can be described by three power laws: two plateaus with exponent zero and one intermediate region with exponent $\gamma$. A functional family that arises frequently in the context of systems that exhibit different power-law regions is the *rational family*. The particular family member we consider is as follows:[4]

$$\hat{\epsilon}(\epsilon_{np}, d \mid l, w, n) = \epsilon_{np} \left\| \frac{d - jp \left( \frac{\epsilon^{\uparrow}}{\epsilon_{np}} \right)^{\frac{1}{\gamma}}}{d - jp} \right\|^{\gamma} \quad \text{where } j = \sqrt{-1} \tag{1}$$

This function's shape is controlled by $\epsilon_{np}$, $\epsilon^{\uparrow}$, $\gamma$, and $p$ (visualized in Figure 1, right). $\epsilon_{np}$ and $\epsilon^{\uparrow}$ are the values of the low and high-error plateaus. $\gamma$ is the slope of the power-law region on logarithmic axes. $p$ controls the density at which the high-error plateau transitions to the power-law region.

**Fitting.** To fit $\hat{\epsilon}(\epsilon_{np}, d \mid l, w, n)$ to the actual data $\epsilon(d, l, w, n)$, we estimate values for the free parameters $\epsilon^{\uparrow}$, $\gamma$, and $p$ by minimizing the relative error $\delta \triangleq \frac{\hat{\epsilon}(\epsilon_{np}, d \mid l, w, n) - \epsilon(d, l, w, n)}{\epsilon(d, l, w, n)}$ using least squares regression. The fit is performed separately for each configuration $(l, w, n)$ for all 30–40 densities, resulting in per-configuration estimates of $\hat{\epsilon}^{\uparrow}$, $\hat{\gamma}$, and $\hat{p}$.

**Evaluating fit.** For a qualitative view,[5] we plot the actual error[6] $\epsilon(d, l, w, n)$ and the estimated error $\hat{\epsilon}(\epsilon_{np}, d \mid l, w, n)$ as a function density for CIFAR-10 ResNets of varying widths (Figure 2, left). Our

---

[4]The expression $\left\| \frac{d - ja}{d - jb} \right\|^{\gamma} = \left( \frac{d^2 + a^2}{d^2 + b^2} \right)^{\frac{\gamma}{2}}$ meaning Eq. 1 can be rewritten as $\epsilon_{np} \left( \frac{d^2 + p^2 \left( \epsilon^{\uparrow} / \epsilon_{np} \right)^{2/\gamma}}{d^2 + p^2} \right)^{\gamma/2}$

[5]Since the error is a 4-dimensional function, projections of it yield qualitative analysis—see Appendix D.
[6]We compute the error as the mean across three replicates with different random seeds and dataset subsamples.

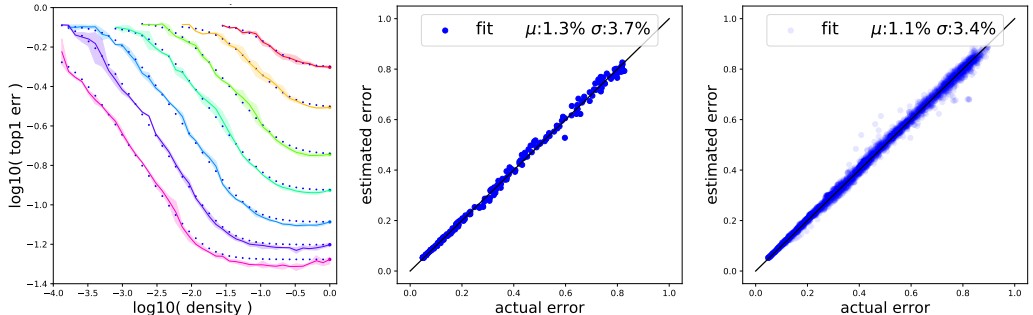

Figure 2: Estimated (blue dots) and actual error (solid lines) when pruning CIFAR-10 ResNets; $w$ varies, $l = 20$, $n = N$ (left). Estimated versus actual error for the same networks (center). Estimated versus actual error for all CIFAR-10 ResNet configurations (right).

Figure 3: Projections of $\epsilon(d, l, w, n)$ onto two-dimensional planes for the CIFAR-10 ResNets, showing contours of constant error. For low enough densities, the contours have linear slopes on the logarithmic axes—depicted by a reference black-dotted line. The density/depth plane (left). The density/width plane (right).

estimated error appears to closely follow the actual error. The most noticeable deviations occur at large densities, where the error dips slightly when pruning whereas we treat it as flat (see Section 6).

Quantitatively, we measure the extent to which estimated error departs from the actual error using the mean $\mu$ and standard deviation $\sigma$ of the relative deviation $\delta$. Figure 2 (center) compares the estimated and actual errors for the networks in Figure 2 (left); Figure 2 (right) shows the same comparison for all configurations of $l$, $w$, and $n$ on CIFAR-10 and the more than 4000 pruned ResNets that result. The relative deviation on all configurations has mean $\mu < 2\%$ and standard deviation $\sigma < 4\%$; this means that, if the actual error is $10\%$, the estimated error is $9.8 \pm 0.4\%$ ($\hat{\epsilon} = (1 - \delta)\epsilon$).

## 4 JOINTLY MODELING ERROR ACROSS ALL DIMENSIONS

In Section 3, we found a functional form $\hat{\epsilon}(\epsilon_{np}, d \mid l, w, n)$ (Equation 1) that accurately predicts the error when pruning a *specific* member of a network family (with depth $l$ and width $w$) trained with a dataset of size $n$. The parameters governing Equation 1 ($\epsilon_\uparrow$, $p$, and $\gamma$ ) were allowed to vary between different configurations and depend on $l$, $w$, $n$. However, we are interested in a single *joint* scaling law $\hat{\epsilon}(\epsilon_{np}, d, l, w, n)$ that, given the unpruned network error $\epsilon_{np}(l, w, n)$, accurately predicts error across *all* dimensions we consider: all members of a network family that vary in depth and width, all densities, and all dataset sizes. Importantly, the parameters of such a *joint* scaling law must be constants as a function of all dimensions. In this section, we develop this joint scaling law.

**Intuition: the error-preserving invariant.** Our desired scaling law $\hat{\epsilon}(\epsilon_{np}, d, l, w, n)$ will be a four-dimensional function of $d$, $w$, $l$, and $n$. To develop this, we study the interdependence between density and depth or width by examining two-dimensional projections of the actual error $\epsilon(d, l, w, n)$ (Figure 3). These plots display contours of constant error as density and depth or width vary.

Consider the projection onto the plane of density and depth (Figure 3, left). The constant-error contours are linear except for in the densest networks, meaning each contour traces a power-law relationship between $d$ and $l$. In other words, we can describe all combinations of densities and widths that produce error $\epsilon_v$ using $l^\phi d = v$, where $v$ is a constant at which network error is $\epsilon_v$ and $\phi$

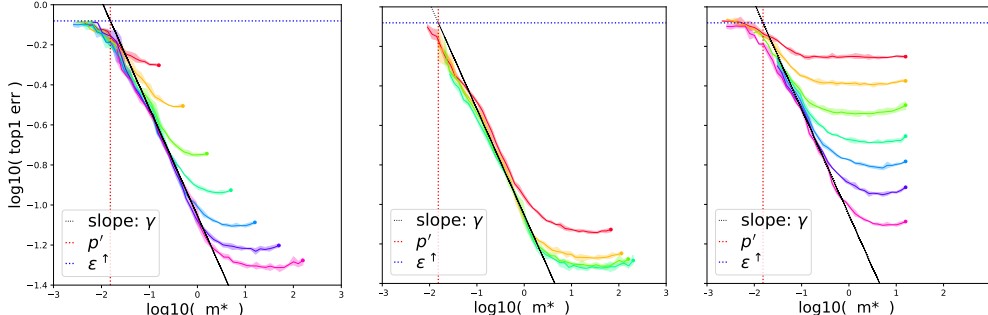

Figure 4: Relationship between $m^*$ and error when pruning CIFAR-10 ResNets and varying $w$ (left, $l = 20$, $n = N$), $l$ (center, $w = 1$, $n = N$), $n$ (right, $l = 20$, $w = 1$). $\gamma$, $\epsilon^\uparrow$, and $p'$ are annotated.

is the slope of the contour on the logarithmic axes. The contours of density and width also have this pattern (Figure 3, right), meaning we can describe a similar relationship $w^\psi d = v'$. Finally, we can combine these observations about depth and width into the expression $l^\phi w^\psi d = v''$.

We refer to the expression $l^\phi w^\psi d$ as the *error-preserving invariant*, and we denote it $m^*$. This invariant captures the observation that there exist many interchangeable combinations of depth, width, and density that achieve the same error and tells us which combinations do so. For example, networks of vastly different densities reach the same error if we vary $l$ and $w$ according to the invariant.

**Functional form.** The invariant allows us to convert the functional form $\hat\epsilon(\epsilon_{np}, d \mid l, w, n)$ for a specific $l$, $w$, and $n$ from Section 3 into a joint functional form $\hat\epsilon(\epsilon_{np}, d, l, w, n)$ for all $l$, $w$, and $n$. Rewriting the definition of the invariant, $d = \frac{m^*}{l^\phi w^\psi}$. We can substitute this for $d$ in the functional form from Section 3. Finally, by rewriting $p$ as $\frac{p'}{l^\phi w^\psi}$ and canceling, we arrive at the expression:

$$\hat\epsilon(\epsilon_{np}, d \mid l, w, n) = \epsilon_{np} \left\| \frac{m^* - jp'\left(\frac{\epsilon^\uparrow}{\epsilon_{np}}\right)^{\frac{1}{\gamma}}}{m^* - jp'} \right\|^\gamma = \epsilon_{np} \left\| \frac{l^\phi w^\psi d - jp'\left(\frac{\epsilon^\uparrow}{\epsilon_{np}}\right)^{\frac{1}{\gamma}}}{l^\phi w^\psi d - jp'} \right\|^\gamma = \hat\epsilon(\epsilon_{np}, d, l, w, n) \quad (2)$$

which is the joint functional form $\hat\epsilon(\epsilon_{np}, d, l, w, n)$ of all four dimensions $d$, $l$, $w$, and $n$. Critically, for this to be a useful joint form, the free parameters $e^\uparrow$, $p'$, and $\gamma$ must be constants shared across all possible values of $d$, $l$, $w$, and $n$. We will assume this is the case and directly quantify how well this assumption holds in the fit section to follow. To glean some qualitative intuition as to why this may be a reasonable assumption, we can examine the relationship between $m^*$ and the generalization error of pruned networks as we vary depth, width, and dataset size (Figure 4). Across all projections, the annotated $e^\uparrow$ (error of the high-error plateau), $\gamma$ (slope of the power-law region) and $p'$ (value of $m^*$ where the high-error plateau transitions to the power-law region) are the same. Note that in Eq. 2 the dependence on $n$ is implicit, through $\epsilon_{np}$. We retain the explicit form $\hat\epsilon(..., n)$ to stress that the lack of explicit dependency on $n$ is non-trivial and was not known prior to our work.

**Fitting.** To fit $\hat\epsilon(\epsilon_{np}, d, l, w, n)$ to the actual data $\epsilon(d, l, w, n)$, we estimate values for the free parameters $\epsilon^\uparrow, \gamma, p', \phi$ and $\psi$ by minimizing the relative error $\delta \triangleq \frac{\hat\epsilon(\epsilon_{np}, d, l, w, n) - \epsilon(d, l, w, n)}{\epsilon(d, l, w, n)}$ using least squares regression. The fit is performed jointly over all configurations of $d$, $l$, $w$, and $n$, resulting in joint estimates of $\hat\epsilon^\uparrow, \hat\gamma, \hat p, \hat\phi,$ and $\hat\psi$. One can also perform a partial fit for a subset of dimensions (e.g., just $d$, $l$, and $n$) by omitting $\phi$ and/or $\psi$ (see Appendix D).

**Evaluating fit.** In Figure 5, we plot the actual error $\epsilon(d, l, w, n)$ and the estimated error $\hat\epsilon(\epsilon_{np}, d, l, w, n)$ for the CIFAR-10 ResNets and ImageNet ResNets (single depth). As in Section 3, our estimated error appears to closely follow the actual error. Deviations arise mainly at high densities where error dips below $\epsilon_{np}$ and low densities approaching high error saturation.

We again quantify the fit of the estimated error using the mean $\mu$ and standard deviation $\sigma$ of the relative deviation $\delta$. The relative deviation on the joint scaling laws for the CIFAR-10 and Imagenet networks has a mean $\mu < 2\%$ and standard deviation of $\sigma < 6\%$.

To contextualize these results, Figure 5 (right) quantifies the variation in error we see over multiple trials of the CIFAR-10 experiments due to using different random seeds. It plots the minimum,

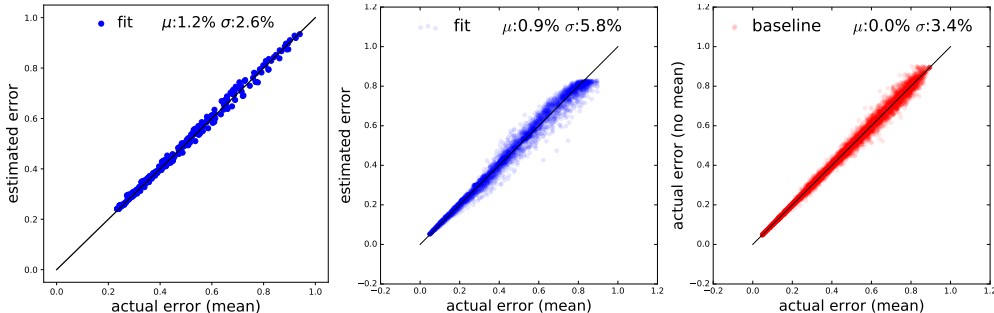

Figure 5: Estimated versus mean actual error for all configurations $(d, w, n)$ for ImageNet (left) and $(d, w, l, n)$ for CIFAR-10 (center). The variation in error when running the same experiment on CIFAR-10 three times with different random seeds (right).

maximum, and mean errors across the three trials we ran.[7] The variation across trials has a standard deviation of $\sigma = 3.4\%$, sizeable relative to the estimation error of $\sigma = 5.8\%$ for the joint scaling law. This indicates that a significant portion of our error may stem from measurement noise.

The functional form has just five parameters and obtains an accurate fit on over 4000 points, suggesting it is a good approximation. In Appendix E, we show that it achieves a similarly good fit for VGG-style networks and for the SynFlow pruning algorithm. In Section 5, we show that it is possible to get a good fit with far fewer points and that the fit has low sensitivity to the choice of points.

## 5 ANALYZING THE SENSITIVITY OF THE FIT TO NUMBER OF POINTS

In Section 4, we showed that our scaling law was accurate when we fit it on all of the available data. Now that we possess the functional form and know that it can accurately model the behavior of IMP, we study the amount of data necessary to obtain a stable,[8] accurate fit. This question is especially relevant when the functional form is applied to new settings—new networks, datasets, or pruning algorithms—and we must collect new data to do so. The functional form has only five parameters, suggesting that few experiments will be necessary.

**Experiments.** To evaluate the effect of the number of points on the stability and accuracy of the fit, we randomly sample varying numbers of points, fit the scaling law to those points, and evaluate the quality of the fit over all points. We sample these points in two ways.

*Experiment 1.* Randomly sample $T$ network configurations $(w, l, n, d)$. This experiment captures the use case of algorithms such as SynFlow (Appendix E), where obtaining data at any density $d$ relies only on possessing the unpruned network, not other densities $d' > d$.

*Experiment 2.* Randomly sample $T$ network configurations $(w, l, n)$ and include all densities $d$ for each configuration. This experiment captures the use case of algorithms such as IMP, where obtaining data at density $d$ requires obtaining all densities $d' > d$. As such, we anticipate that data will be obtained by iteratively pruning a small number of configurations $(w, l, n)$ to low density.

**Results.** We perform each experiment for many different values of $T$ on the CIFAR-10 ResNets pruned with IMP. We repeat the experiment at each value of $T$ 30 times (with a different sample of points) and report the mean and standard deviation of $\mu$ and $\sigma$ for the fit. Experiments 1 and 2 appear in Figure 6. The shaded areas represent one standard deviation from the mean in each direction. On Experiment 1, when just 40 configurations of $(w, l, d, n)$ are available, the standard deviation on both $\mu$ and $\sigma$ is just one percentage point. On Experiment 2, when just 15 random configurations of $(w, l, n)$ are available at all densities, we similarly achieve standard deviation below 1%. In both cases, as the number of networks increases, the standard deviation decreases further.

---

[7] We only ran a single trial of the ImageNet experiments due to the significant cost of collecting data.

[8] Stability is defined as a small change in output relative to a change in input. The requirement here is that a change in choice of points leads to a small expected change in estimation accuracy in expectation.

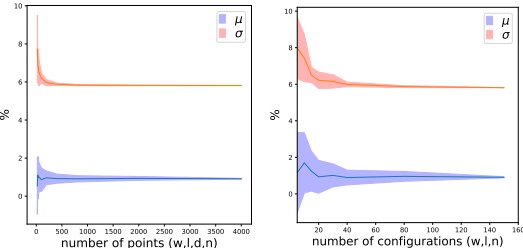

Figure 6: The effect of the number of points used to fit our scaling law (on the CIFAR-10 ResNets pruned with IMP) on $\mu$ and $\sigma$. Left: experiment 1 from Section 5 (random points $w$, $l$, $d$, $n$). Right: experiment 2 from Section 5 (random configurations $w$, $l$, $n$ and all densities).

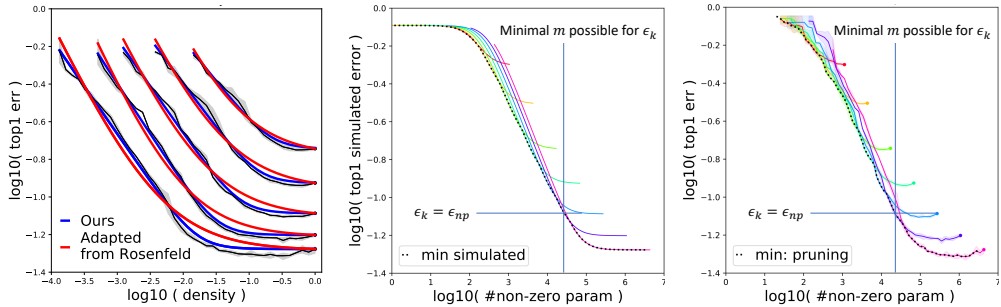

Figure 7: Error of the CIFAR-10 ResNets as width varies (left); actual error (black), error of our scaling law (blue), and error of the scaling law adapted from Rosenfeld et al. (2020) (red). Estimated error from our scaling law as width varies for the CIFAR-10 ResNets (center); in dotted black is the minimal number of parameters for each error $\epsilon_k$. Same as center but using actual error (right).

These results show that, now that our scaling law is known, it is possible to obtain an accurate (and stable) estimation using far less data than we used to evaluate the quality of the fit in the main body of the paper. Importantly, the experiments we perform in this section are particularly naive. We make no effort to ensure that the configurations we select represent a diverse range of widths, depths, dataset sizes, and densities. By selecting these configurations in a strategic way, we believe it would be possible to further reduce the number of configurations necessary to obtain a similarly accurate fit.

## 6   PRINCIPLES FOR SELECTING A FUNCTIONAL FAMILY

In this section, we discuss some of the key criteria that led us to select this particular functional form and opportunities for further refinement.

**Criterion 1: Transitions.** In Section 3, we observe that, when pruning a neural network, error has a low-error plateau, a power-law region, and a high-error plateau. Between these regions are *transitions* where error varies smoothly from one region to the next. Matching the shape of these transitions was a key consideration for selecting our function family. To illustrate the importance of properly fitting the transitions, Figure 7 (left) shows two possible functional families for fitting the relationship between density and error for CIFAR-10 ResNets. Actual error is in gray, and the functional form from Section 3 is in blue. In red is the fit for a functional form adapted from the one that Rosenfeld et al. (2020) use to model the relationship between width and error. The difference between these functional families is the way they model transitions, and the one we choose in this paper better models the transitions in our setting. For further discussion of this comparison, see Appendix G.

**Criterion 2: A small number of interpretable parameters.** Selecting a functional form is not merely a curve-fitting exercise. We seek the underlying structure that governs the relationships between $d, l, w, n$, and error in a manner akin to a law of physics. As such, our functional form should have a small number of parameters that are *interpretable*. In our functional form (Equation 2), each parameter has a clear meaning. The parameters $\epsilon^\uparrow$, $p'$, and $\gamma$ control the high-error plateau, the transition to the power-law region, and the slope of the power-law region. $\phi$ and $\psi$ control the interchangeability of width and depth with density. We approximate error over multiple orders

of magnitude and over 4,000 configurations of ResNet-20 on CIFAR-10 with just five parameters, indicating we have distilled key information about the behavior of pruning into our functional form.

**Sources of systemic error and limitations of our approximation form.** By seeking to minimize the number of parameters in our functional form, we leave some phenomena unmodeled. In particular, there are two phenomena we have chosen *not* to model that introduce systemic error. First, the low-error plateau is not a plateau. Error often improves slightly at high densities before returning to $\epsilon_{np}$ during the transition to the power-law region. Our model treats the region as flat and treats error as monotonically increasing as density decreases. This source of error accounts for a bias of $\sim 1\%$ relative error in our estimation (Appendix H). Second, we model both transitions (between the power-law region and each plateau) with a single shape and the same transition rate. If we treated each transition separately and used higher-order terms in the rational form, we could potentially reduce some of the residual error in our estimation at the cost of additional complexity.

## 7  IMPLICATIONS AND CONCLUSIONS

Our main contribution is a functional form $\hat{\epsilon}(\epsilon_{np}, d, l, w, n)$ that accurately predicts the error when pruning members of a network family using both IMP and SynFlow. There are several broader implications of our ability to characterize pruning in this way. The mere existence of this functional form means there is indeed structure to the way pruning affects error. Although prior work (Cai et al., 2019) has implicitly relied on such structure, we are the first to explicitly describe it. This functional form enables a framework in which we can reason conceptually and analytically about pruning. In doing so, we can make new observations about pruning that are non-obvious or costly to exhaustively demonstrate empirically. For example, recall our motivating question:

*Given a family of neural networks, which should we prune (and by how much) to obtain the network with the smallest parameter-count such that its error does not exceed some threshold $\epsilon_k$?*

This is an optimization problem—find the configuration of $d$, $l$, and $w$ that minimizes parameter-count subject to an error constraint: $\mathrm{argmin}_{w,l,d}\, m$ s.t. $\hat{\epsilon} = \epsilon_k$. For ResNets $m \propto dlw^2$, yielding:

$$l, w, d = \operatorname*{argmin}_{l,w,d} lw^2 d \quad \text{s.t.} \quad \epsilon_{np} \left\| \left[ l^\phi w^\psi d - jp'(\epsilon^\uparrow / \epsilon_{np})^{1/\gamma} \right] \cdot \left[ l^\phi w^\psi d - jp' \right]^{-1} \right\|^\gamma = \epsilon_k$$

which is solvable directly without running any further experiments.

Using this approach, we can derive a useful insight. In the pruning literature, it is typical to report the minimum density at which the pruned network can match the error $\epsilon_{np}(l, w)$ of the unpruned network (Han et al., 2015). However, our scaling law suggests that this is not the smallest model that achieves error $\epsilon_{np}(l, w)$. Instead, it is better to train a larger network with depth $l'$ and width $w'$ and prune until error reaches $\epsilon_{np}(l, w)$, even as that results in error well above $\epsilon_{np}(l', w')$. This analytic result parallels and extends the findings of Li et al. (2020) on NLP tasks.

Figure 7 (center) illustrates this behavior: it shows the error predicted by our scaling law for CIFAR-10 ResNets with varying widths. The dotted black line shows the minimal parameter-count at which we predict it is possible to achieve each error. Importantly, none of the low-error plateaus intersect this black dotted line, meaning a model cannot be minimal until it has been pruned to the point where it increases in error. This occurs because the transitions of our functional form are gradual. On the other hand, if we start with a model that is too large, it will no longer be on the black line when it has been pruned to the point where its error reaches $\epsilon_{np}(l, w)$.[9] In Figure 7 (right), we plot the same information from the actual CIFAR-10 data and see the same phenomena occur in practice. The difference between the estimated and actual optimal parameter count is no more than 25%.

Looking ahead, there are several opportunities for future work. Better understanding the sources of systematic error (error dips and transition shape) is a promising avenue for making it possible to *extrapolate* from small-scale settings to large-scale settings (see Appendix F for a forward looking discussion of extrapolation). Furthermore, although we focus on pruning for image classification and networks and pruning methods differ in other contexts (e.g., NLP), the generality of our functional form to different pruning strategies and network families suggests it may have broader applicability.

---

[9]This behavior occurs since $m \not\propto m^*$ for IMP. Interestingly, for SynFlow $m \propto m^*$ such that sufficiently large networks are equivalent.

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

CONTENTS OF APPENDICES

## A    PRUNING ALGORITHMS

### A.1    FORMAL STATEMENT OF ITERATIVE MAGNITUDE PRUNING (IMP)

---
**Algorithm 1** Iterative Magnitude Pruning (IMP) with weight rewinding to epoch 10 and $N$ iterations.

---
1: Create a neural network with randomly initialized weights $W_0 \in \mathbb{R}^d$ and initial pruning mask $m = 1^d$
2: Train $W_0$ to epoch 10, resulting in weights $W_{10}$
3: **for** $n \in \{1, \ldots, N\}$ **do**
4:     Train $m \odot W_{10}$ (the element-wise product of $m$ and $W_{10}$) to final epoch $T$ and weights $m \odot W_{T,n}$
5:     Prune the 20% of weights in $m \odot W_{T,n}$ with the lowest magnitudes. $m[i] = 0$ if $W_{T,n}[i]$ is pruned
6: Return $m$ and $W_{T,n}$

---

### A.2    SYNFLOW

Unlike IMP, SynFlow is a pruning algorithm that prunes neural networks *before* any training has taken place (Tanaka et al., 2020). To do so, SynFlow computes the "synaptic strengths" of each connection and prunes those weights with the lowest synaptic strengths (see Algorithm 2 below for the details on computing the synaptic strengths).

Importantly, SynFlow prunes iteratively. It prunes a small number of weights, recalculates the synaptic strengths once those weights have been fixed to zero, and then prunes again. To prune to sparsity $s$, SynFlow iteratively prunes from sparsity $s^{\frac{n-1}{100}}$ to sparsity $s^{\frac{n}{100}}$ for $n \in \{1, \ldots, 100\}$.

After pruning, SynFlow trains the network normally using the standard hyperparameters. SynFlow computes the synaptic strengths as follows:[10]

---
**Algorithm 2** Computing the synaptic strengths of a network with weights $W$.

---
1: Replace all weights $w \in W$ with their magnitudes $|w|$.
2: Forward propagate an input of all 1's
3: Take the sum of the logits $R$.
4: The synaptic strength for each weight $w$ is the gradient $\frac{dR}{dw}$.

---

Note that this algorithm leads to exploding activations on deeper networks, so we do not vary network depths in any of our experiments involving SynFlow.

---
[10]https://github.com/ganguli-lab/Synaptic-Flow

## B Experimental Details

### B.1 ResNets

We study the residual networks (ResNets) designed by He et al. (2016) for CIFAR-10 and ImageNet. ResNets for CIFAR-10 are composed of an initial convolutional layer, three sets of $B$ residual blocks (each with two convolutional layers and a skip connection), and a linear output layer. The sets of blocks have 16, 32, and 64 convolutional channels, respectively.

ResNets for ImageNet are composed of an initial convolutional layer, a max-pooling layer, four sets of residual blocks (each with three convolutional layers and a skip connection), and a linear output layer. The sets of blocks have 64, 128, 256, and 512 convolutional channels, respectively.

We place batch normalization before the ReLU activations.

To vary the width of the networks, we multiply the number of convolutional channels by the width scaling factor $w$. To vary the depth of the CIFAR-10 ResNets, we vary the value of $B$. The depth $l$ of the network is the total number of the layers in the network, not counting skip connections.

### B.2 VGG Networks

We study the VGG-16 variant of the VGG networks for CIFAR-10 as provided by the OpenLTH repository.[11] The network is divided into five sections, each of which is followed by max pooling with kernel size 2 and stride 2. The sections contain 3x3 convolutional layers arranged as follows:

| Section | Width | Layers |
|---|---|---|
| 1 | 64 | 2 |
| 2 | 128 | 2 |
| 3 | 256 | 3 |
| 4 | 512 | 3 |
| 5 | 512 | 3 |

The network has ReLU activations and batch normalization before each activation. To vary the width of VGG-16, we multiply each of the per-segment widths by the width scaling factor $w$.

When pruning VGG, we consider the following configurations:

| Network Family | $N_{\text{train}}$ | $N_{\text{test}}$ | Densities ($d$) | Depths ($l$) | Width Scalings ($w$) | Subsample Sizes ($n$) |
|---|---|---|---|---|---|---|
| CIFAR-10 VGG-16 | 50K | 10K | $0.8^i, i \in \{0, \ldots, 37\}$ | 16 | $2^i, i \in \{-4, \ldots, 0\}$ | $\frac{N}{i}, i \in \{1\}$ |

### B.3 Training Hyperparameters

We train CIFAR-10 ResNets and VGG-16 for 160 epochs with a batch size of 128. The initial learning rate is 0.1, and it drops by an order of magnitude at epochs 80 and 120. We optimize using SGD with momentum (0.9). We initialize with He uniform initialization. Data is augmented by normalizing, randomly flipping left and right, and randomly shifting by up to four pixels in any direction (and cropping afterwards). All CIFAR-10 networks are trained on GPUs.

We train ImageNet ResNets for 90 epochs with a batch size of 1024. The initial learning rate is 0.4, and it drops by an order of magnitude at epochs 30, 60, and 80. We perform linear learning rate warmup from 0 to 0.4 over the first 5 epochs. We optimize using SGD with momentum (0.9). We initialize with He uniform initialization. Data is augmented by normalizing, randomly flipping left and right, selecting a random aspect ratio between 0.8 and 1.25, selecting a random scaling factor between 0.1 and 1.0, and cropping accordingly. All ImageNet networks are trained on GPUs.

---

[11]github.com/facebookresearch/open_lth

## C FULL DATA FOR KEY OBSERVATIONS IN SECTION 3

In this appendix, we show that our observations from Section 3 hold when varying all dimensions (depth, width, and dataset size) on both the CIFAR-10 and ImageNet ResNets for IMP. Figure 8 shows the error versus density when changing width (left) depth (center) and data (right). In Figure 9, we similarly show the dependency of the error on density for Imagenet when varying width (left) and dataset size (right).

In Figure 8, we observe that all curves have a similar slope in the power-law region. In Equation 1, this implies that while $\gamma$ is allowed to vary with $l$, $w$ and $n$, it is in practice approximately a constant. Similarly, the high-error plateau $\epsilon^{\uparrow}$ is also shared across curves such that it too is approximately constant. In contrast, the transition from high-error plateau to the power-law region is not constant as a function of density. Section 4 finds exactly this dependency of the transition parameter $p$.

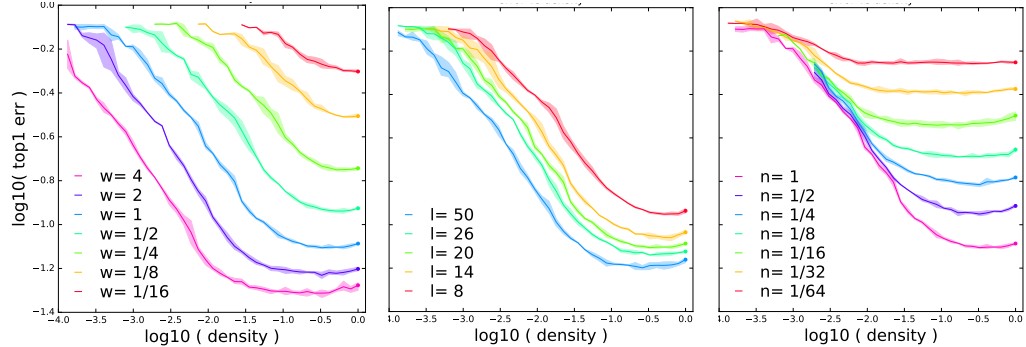

Figure 8: Relationship between density and error when pruning CIFAR-10 ResNets and varying $w$ (left, $l = 20$, $n = N$), $l$ (center, $w = 1$, $n = N$), $n$ (right, $l = 20$, $w = 1$)

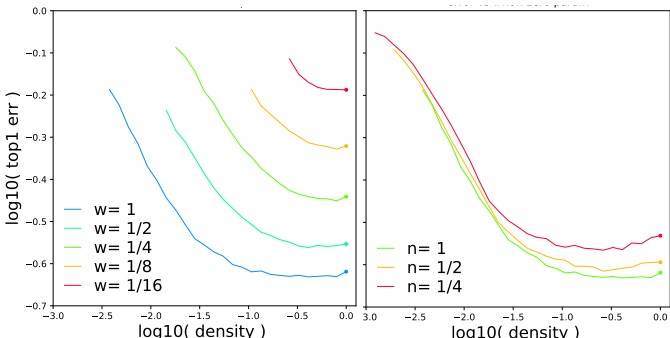

Figure 9: Relationship between density and error when pruning Imagenet ResNet-50 and varying $w$ (left, $n = N$), and $n$ (right, $w = 1$)

# D    PARTIAL (PROJECTIONS) FIT RESULTS FOR SECTION 4

In Section 4, we fit the error jointly as a function of all dimensions showing that Equation 2 provides a good approximation to the error in practice. In this appendix, we consider important sub-cases, such as the case when one wishes to scale only one degree of freedom while pruning. This serves both a practical scenario, but also allows for a qualitative visualization of the fit (and typical sources of error), which is otherwise difficult to perform over all dimensions jointly. From a practical standpoint, in this case one need not estimate the parameters associated with the fixed degree of freedom.

Recall that, given the non-pruned network error $\epsilon_{np}$, all dependencies on the individual structural degrees of freedom $l, w$ are captured by the invariant $m^* \triangleq l^\phi w^\psi d$. This means that, if one wishes to estimate the error while pruning when holding width fixed, we need not estimate $\psi$. Similarly if depth is held constant, we need not estimate $\phi$.

Figure 10 shows these partial fits. Shown from left to right are the fits done while pruning and varying width, depth and data respectively. Correspondingly, these fits omit separately $\psi$ or $\phi$ or omit both when depth nor width are scaled. The fits were performed with all available density points for each dimension. For CIFAR-10: 7 widths, 224 points for the width partial fit; 7 dataset fractions, 240 points for the data partial fit; 4 depths, 164 points for the depth partial fit. For ImageNet: 5 widths, 83 points for the width partial fit; 3 dataset fractions, 86 points for the data partial fit.

This exercise, apart from its practical implications, highlights the fact that there are in effect two groups of parameters comprising the estimation. The first are the parameters $\epsilon^\uparrow$, $\gamma$ and $p'$ which control the dependency as a function of density (or more generally, as a function of the invariant). The second are $\phi$ and $\psi$ which are properties of the architectural degrees of freedom captured by the invariant. Moreover, within the first group of parameters $\epsilon^\uparrow$, $\gamma$, can be isolated and found from a single pruning curve, as they are not a function of $l, w, n$.

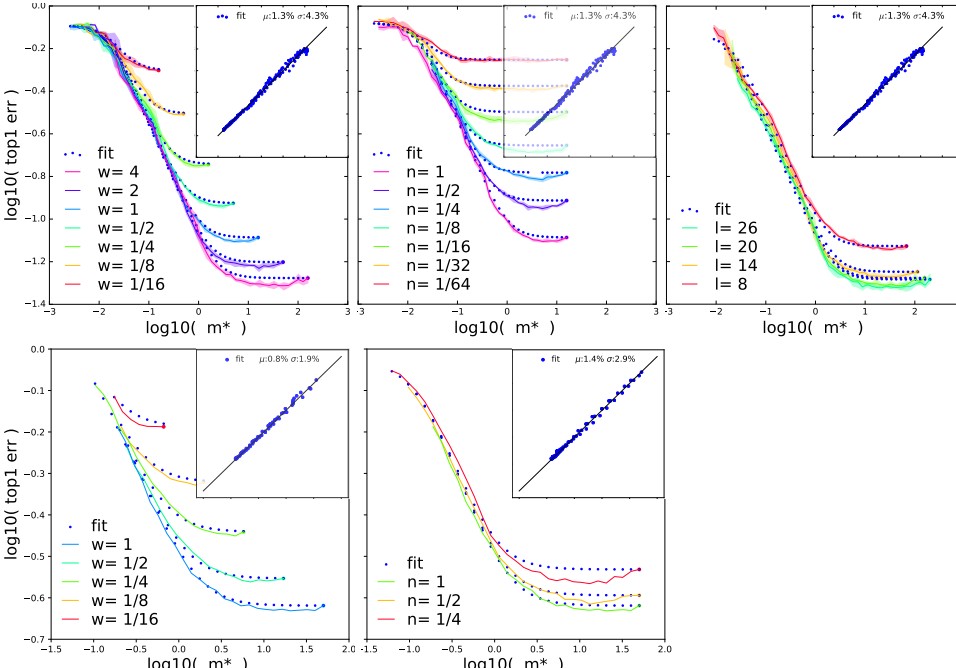

Figure 10: Top row: CIFAR-10. Bottom row: ImageNet. Left: varying width. Center: varying dataset size. Right: varying depth. Lines are the actual error and dots are the estimated error.

# E ADDITIONAL PRUNING ALGORITHMS AND ARCHITECTURES

In this appendix, we show that our functional form applies to both an additional network architecture (VGG-16 on CIFAR-10) and an additional pruning algorithm (SynFlow). We add these additional comparisons in the following Figures:

- Figure 11: VGG-16 on CIFAR-10 with IMP as width varies. $\mu < 3\%$, $\sigma < 7\%$. Notably, the measurement error in this case is large ($\sigma \sim 12\%$), dominating (over the approximation error) the total fit error. The fit averages out some of this error, resulting in a fit error which is lower than the measurement error.
- Figure 12: ResNet-20 on CIFAR-10 with SynFlow as width varies. $\mu < 1\%$, $\sigma < 4\%$.
- Figure 13: VGG-16 on CIFAR-10 with SynFlow as width varies. $\mu < 1\%$, $\sigma < 4\%$.

Note that SynFlow suffers from exploding activations on deeper networks, so we do not vary ResNet depth in any of our SynFlow experiments.

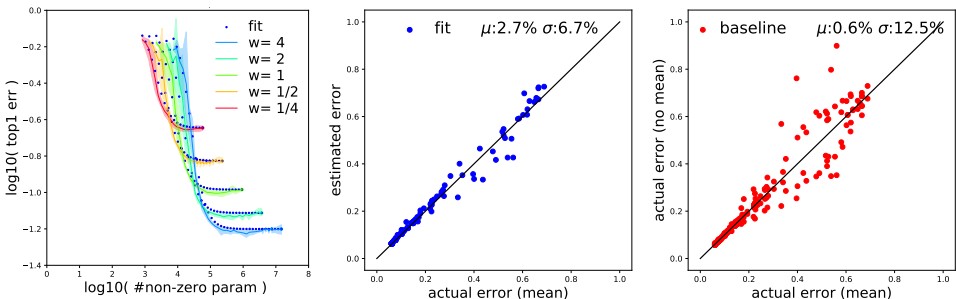

Figure 11: Fit for VGG-16 on CIFAR-10 with IMP pruning.

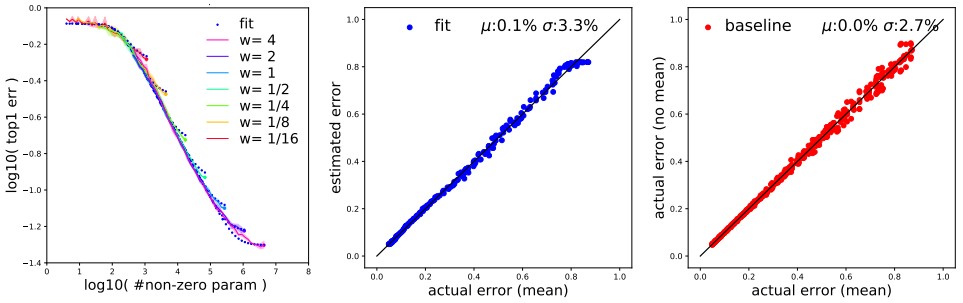

Figure 12: Fit for ResNet-20 on CIFAR-10 with SynFlow pruning.

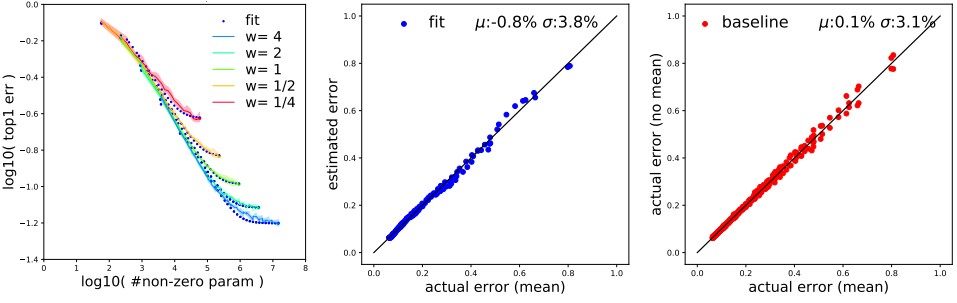

Figure 13: Fit for VGG-16 on CIFAR-10 with SynFlow pruning.

# F    TOWARDS EXTRAPOLATION

**Background.** In the main body, we showed that our scaling law accurately fits the error of pruned neural networks. As, such it has predictive power, allowing us to reason in a principled manner about pruning trade-offs. Similarly, it allows to make predictions about what would happen at larger model and data scales than explored here. Importantly, only a few experiments need be performed to find the coefficients for the scaling law (see Appendix 5).

However, we could ask, how accurately can we estimate the scaling law parameters from even smaller scales? That is, is it possible to fit our scaling law to data from networks with deliberately smaller depths, widths, and dataset sizes and accurately predict the error of larger-scale models? If so, we could make informed decisions about pruning large-scale models through small-scale experiments alone, saving the costs associated with large scale training and pruning.

Outside the context of pruning, the scaling laws of Rosenfeld et al. (2020) (for both language models and image classification) and Kaplan et al. (2020) (for predicting the expected performance of GPT-3 (Brown et al., 2020) at very large scale) have been shown to extrapolate successfully in this manner.

**Results on CIFAR-10.** In Figure 14, we show the result of extrapolating from small-scale networks on CIFAR-10 ($w = \frac{1}{8}, \frac{1}{4}; l = 14, 20$) to all widths and depths on CIFAR-10. Extrapolation prediction is still accurate: $\mu < 7\%$, $\sigma < 6\%$ (vs. $\mu < 1\%$, $\sigma < 6\%$ in the main body).

**Future work.** However, extrapolation is particularly sensitive to systemic errors. Specifically, the transitions and the error dips can lead to large deviations when extrapolating. For ImageNet, the error dips (especially on small dataset sizes) are especially pronounced, preventing stable extrapolation. In order to improve extrapolation performance, future work should explore the challenges we discuss in Section 6: approaches to either model or mitigate these dips and to improve the fit of the transitions.

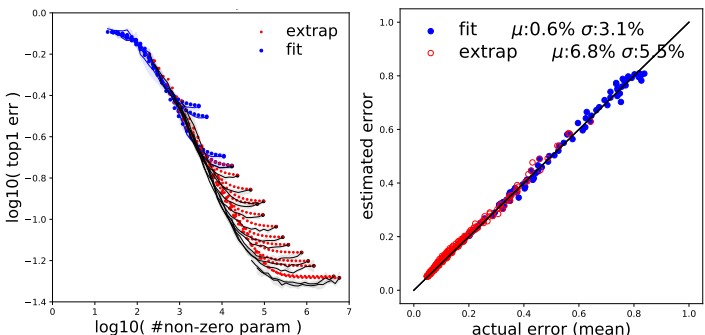

Figure 14: Extrapolation results from four pruned networks on CIFAR10 $w = \frac{1}{8}, \frac{1}{4}; l = 14, 20$ to all larger networks ($n = 1$). Fit results are in blue, extrapolation in red, actual in black. Error versus number of non-zero parameters (left). Estimated versus actual errors (right).

## G   COMPARISON OF PRUNING AND NON-PRUNING SCALING LAWS

In this appendix, we contrast the behavior of the error when pruning with the behavior of the error in the non-pruning setting. Hestness et al. (2017) show the the error follows a saturating power-law form when scaling data (with both low and high-error plateaus) but does not model them. Rosenfeld et al. (2020) unify the dependency on data and model size while approximating the transitions between regions; they propose the following form:

$$\tilde{\epsilon}(m, n) = an^{-\alpha} + bm^{-\beta} + c_{\infty} \tag{3}$$

$$\hat{\epsilon}(m, n) = \epsilon_0 \left\| \frac{\tilde{\epsilon}(m, n)}{\tilde{\epsilon}(m, n) - j\eta} \right\| \tag{4}$$

where $m$ is the total number of parameters and $n$ is the dataset size. $a, b, \alpha, \beta, c_{\infty}$, and $\eta$ are constants, and $\epsilon_0$ plays the role of $\epsilon^{\uparrow}$ in our notation.

Rosenfeld et al. model the upper transition—from power-law region to the high-error plateau—by a rational form in a fashion similar to the approach we take. The key difference is that we consider a power of the polynomials in the numerator and denominator of the rational form, where in Eq. 3 the power is hidden in the term $\tilde{\epsilon}$.

The biggest difference arises when considering the lower transition (between the low-error plateau and the power-law region). This transition is captured by Eq. 3. Considering either the width or depth degrees of freedom $x \in \{w, l\}$, Eq. 3 can be re-written as:

$$\tilde{\epsilon}(x) = b_x x^{-\beta_x} + c_x \tag{5}$$

Where $b_x$ and $\beta_x$ are constants and $c_x$ is a constant as a function of $x$ (it is only a function of the data size $n$).

Figure 15 (right) shows the error versus depth for different dataset sizes. In grey is the actual error, while in red is the best fit when approximating the error by Eq. 5. Qualitatively, one sees that the fit using Eq. 5 does indeed closely match the error in practice.

Recall that we are interested in comparing the errors as a function of the density. A requirement from any functional form used to model the dependency on the density is to degenerate to the error of the non pruned model $\epsilon_{np}$ at $d = 1$. We adapt Eq. 5 by solving the relation between $b_x$ and $c_x$ meeting this constraint, to arrive at:

$$\tilde{\epsilon}(x) = b_x x^{-\beta_x} + \epsilon_{np} - b_x \tag{6}$$

Contrast Eq. 5 with the functional form we propose in Eq. 1, re-written here for convenience:

$$\hat{\epsilon}(d, \epsilon_{np} \mid l, w, n) = \epsilon_{np} \left\| \frac{d - jp \left( \frac{\epsilon^{\uparrow}}{\epsilon_{np}} \right)^{\frac{1}{\gamma}}}{d - jp} \right\|^{\gamma} \quad \text{where } j = \sqrt{-1} \tag{7}$$

This can be simplified to capture only the lower transition—far enough from the upper transition ($d \gg p$)—to:

$$\hat{\epsilon}(d, \epsilon_{np} \mid l, w, n) = \epsilon_{np} \left\| \frac{d - jp \left( \frac{\epsilon^{\uparrow}}{\epsilon_{np}} \right)^{\frac{1}{\gamma}}}{d} \right\|^{\gamma} \tag{8}$$

Figure 15 (left) shows error versus density for different widths. In blue is the fit with Eq. 8 which follows closely the actual error (black) while in red is the fit with Eq. 6 which deviates noticeably in comparison.

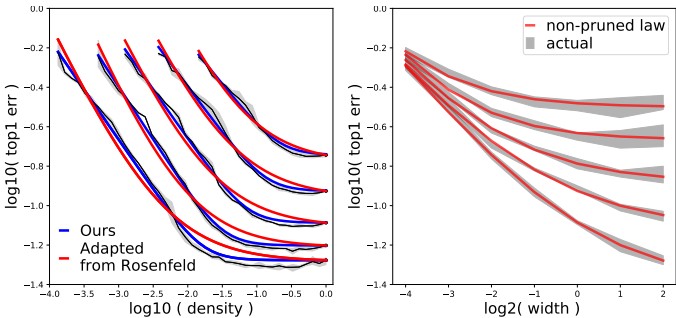

Figure 15: (Left) Error versus density for different widths. In blue is the fit eq. 2 follows closely the actual error (black) while in red is the fit for the adapted from Rosenfeld et al. (2020) which deviates noticeably in comparison. (Right) error of non-pruned networks versus width for different data, fit shown (solid red) for the non-pruning scaling from Rosenfeld et al. (2020).

We have seen that in practice that the form of Eq. 6 does not match well the pruning case, where the mismatch originates from lower transition shape. We have thus reached a phenomenological observation distinguishing the pruning and non-pruning forms; we leave the study of the origins of this phenomenon for future work.

# H    THE EFFECT OF ERROR DIPS ON ESTIMATION BIAS

In this appendix, we consider the effect of the error dips on our estimator as discussed in Section 4. As we mention in that section, when pruning a network, error often dips below $\epsilon_{np}$ during the low-error plateau.

Recall that we find the parameters in our estimator (Equation 2) by minimizing the MSE of relative error $\delta$. Our estimation has bias if $\mathbb{E}\left(\hat{\epsilon} - \epsilon\right) \neq 0$ where the expectation is over all model and data configurations. Equivalently, the relative bias is $\mu \triangleq \mathbb{E}\delta = 0$ iff the estimator is unbiased. The Estimator captured by the joint form in Equation 2 is a monotonically increasing function of the density. It is also constrained such that at density $d = 1$ it is equal to the non-pruned error $\epsilon_{np}$. It thus, can not reduce The MSE to zero, as it can not decrease to match the actual error dips. This results in the bias of the relative error $\mu$ which in practice is $\sim 1\%$.

