# OpenReview forum: "On the Predictability of Pruning Across Scales"
_ICLR.cc/2021/Conference — Reject_

### Official Review · AnonReviewer1 · 2020-10-27

**Rating:** 6
**Confidence:** 3

**Review:**

This paper studies how to estimate the performance of pruned networks using regression models. The authors first empirically observe that there exist three distinct regions of sparsity: (1) In the low-sparsity regime, pruning does not decrease the accuracy (2) In the mid-sparsity regime, a linear relationship between the sparsity and the accuracy is observed (3) In the high-sparsity regime, pruning does not decrease the accuracy again. Under this observation, the authors proposed a regression model called the rational family and empirically verified its performance. The authors further extended this model to incorporate the network width and depth under some empirical observation called the error-preserving invariant. The authors performed experiments to verify different perspectives of the proposed functional form.

Overall, I like the main idea and experiments on the interpolation/extrapolation using the proposed functional form. However, I believe that the following concerns are critical.
- The authors removed a huge amount of test configurations: about 8000 configurations among 12000 configurations are eliminated for CIFAR-10 ResNet. It significantly reduces the reliability of estimations from the proposed functional form as the removed cases are occasionally observed in practice.
- The authors only consider limited experimental setups. For example, the effects of network architectures, rewinding weights, iterative pruning, and training epoch to their observation (e.g., having three different phases, error-preserving invariant) are not verified. I am curious whether the proposed functional form is still valid for other architectures (e.g., DenseNet, recurrent networks, language models) and without rewinding.

---

> ### Author Response · Authors · 2020-11-22
> **Author Response**
>
> We thank the reviewer for the feedback. We have responded to the reviewer’s comments in-line below.
>
> ---
>
> _The authors removed a huge amount of test configurations...It significantly reduce the reliability of the estimations from the proposed functional form as the removed cases are occasionally used in practice._
>
> Of the 12,054 possible CIFAR-10 configurations described in Section 2:
> 1. ~4,000 are not valid networks: they are impossible to train because the network is so heavily pruned that it has become disconnected.
> 2. ~4,000 represent configurations of the network that would not be useful in practice because a shallower or less wide version performed better. These configurations occur predominantly when we use small subsamples of the dataset; of the 144 configurations of _l_, _w_, and _n_ that we eliminate, only 6 occur on the full dataset.
>
> ---
>
> _The authors only consider limited experimental setups….I am curious whether the proposed functional form is still valid for other architectures (e.g., DenseNet, recurrent networks, language models) and without rewinding._
>
> In Appendix E, we explore additional configurations: the VGG network family and the SynFlow pruning algorithm (which does not involve rewinding).
>
> In order to ensure our scaling law obtains an accurate fit across orders of magnitude (and to evaluate this fit across multiple replicates), adding a new architecture or dataset requires training an immense number of networks. We pushed our resources and our budget to the limit in order to obtain the results in the current paper.
>
> We agree that it will be exciting to explore this scaling law across additional settings and architectures, but we believe the current experiments are convincing and comprehensive and we lack the resources to add more large-scale settings.

---

### Official Review · AnonReviewer4 · 2020-10-28
**On the Predictability of Pruning Across Scales**

**Rating:** 6
**Confidence:** 4

**Review:**

Paper summary

The authors propose a functional approximation to the error of pruned convolutional neural networks as a function of network hyperparameters. This functional approximation depends on a number of hyperparameters that are fit on the error of already trained and pruned networks on a certain task (in this case, image classification on CIFAR-10 and ImageNet are the tasks under consideration). The authors demonstrate that this fit is very accurate over many orders of magnitude, which demonstrates their hypothesis on the power law nature of the error distribution as a function of the hyperparameters under consideration.

---------------------------------------------------------------------------------------------------------------------------
Positives and negatives

+The paper is very well written. It is very easy to follow the author’s argument and the figures illustrate the point in a concise manner.
+The experimental section is very strong. The authors do a good job of motivating their hypothesis with empirical data, and explaining why they propose a power law.
+After proposing the power law hypothesis, the authors fit the expression on a large amount of data, and convincingly show that the power law expression holds over multiple orders of magnitude.
-The contribution of this paper is not as significant as I was lead to believe after reading the title and the abstract. While the authors have convinced me that the behavior of a network after pruning under different densities does follow a power law, the expression they derive depends on fitting 5 hyperparameters (equation 2), and after the fit can only be used only for that specific architecture and dataset.
-The generalization of this expression is not well explored. The authors only consider one type of CNN and two datasets. Trying different CNN architectures and datasets (small datasets such as SVHN, mnist or fashion-mnist would have been fine, as long as quite different architecture types would have been probed).

---------------------------------------------------------------------------------------------------------------------------
Recommendation

I recommend a weak accept for this paper, in light of the following considerations: the paper is well written and it provides an interesting insight (the power law structure of the error as networks are pruned), the paper’s formula could be useful to applied practitioners. However, I believe the way the results are presented are strongly overstated. I would like to see the abstract changed to reflect that the functional approximation derived is an empirical fit dependent on architecture and task; and I would like to see a little more variation in network architectures investigated. If those issues are addressed I would be willing to upgrade my recommendation.

---------------------------------------------------------------------------------------------------------------------------
Questions

* I am a bit confused as to why n is a hyperparameter in eq 2 at all as it only seems to feature indirectly via the error. Could you add a bit more explanation as to why that is considered?
* Given that the formula is fitted with empirical data I find it surprising that the study of fit quality as a function of data points is relegated to the appendix. I think this point is important enough to feature in the main text, and would like to see results as the number of points trends towards 5 (the minimum number of points for which we’d expect a consistent fit). Maybe a plot with fit residual as a function of fit points?

---------------------------------------------------------------------------------------------------------------------------
Feedback (not related to the score)

* Figure 3 is not very informative. I guess the point is to compare the quality of the fit contours with the empirical data. In that case the information about the fit is not enough ( a single line). I would rework the figure to contain full iso-lines for both fit and empirical data. And the caption could be more informative too (why do we need to see the density/depth plane and density/width plane if they are almost identical)?
* In Eq. 1, it would have been useful if p had been introduced in the notation section.
* In figure 4, please add legends to the different colors in each plot.

---

> ### Author Response · Authors · 2020-11-22
> **Author Response**
>
> We thank the reviewer for the detailed feedback. We have responded to the reviewer’s comments in-line below.
>
> ---
>
> _The contribution of this paper was not as strong as I was lead to believe after reading the title and abstract. While the authors have convinced me that the behavior of a network after pruning under different densities does follow a power law, the expression they derive depends on fitting 5 hyperparameters, and after the fit can only be used for the specific architecture and dataset._
>
> We have revised the abstract according to your feedback, and we would appreciate any further feedback you have to ensure that readers are clear on our contributions.
>
> ---
>
> _The generalization of this expression is not well explored. The authors only consider one type of CNN and two datasets._
>
> In Appendix E, we explore additional configurations: the VGG network family and the SynFlow pruning algorithm.
>
> In order to ensure our scaling law obtains an accurate fit across orders of magnitude (and to evaluate this fit across multiple replicates), adding a new architecture or dataset requires training an immense number of networks. We pushed our resources and our budget to the limit in order to obtain the results in the current paper.
>
> We agree that it will be exciting to explore this scaling law across additional settings and architectures. We lack the resources to add more large-scale settings and we were not able to run additional experiments during the rebuttal period, but we will explore adding more smaller-scale settings (such as MNIST and SVHN as you suggest) for the next version of the paper.
>
> ---
>
> _I am a bit confused as to why n is a hyperparameter in eq 2 at all as it only seems to feature indirectly in the error. Could you add a bit more explanation as to why that is considered?_
>
> Although we find that $n$ only features indirectly in the error, prior to our work it was not known how the error of pruned networks depended on dataset size. We continue to explicitly denote the dependency of the generalization error under pruning on $n$ to stress that it is generally a function of dataset size even though this notation is redundant in hind-sight given our findings. We have added a clarification of this point in the paper.
>
> ---
>
> _Given that the formula is fitted with empirical data I find it surprising that the study of fit quality as a function of data points is relegated to the appendix._
>
> We have moved this Appendix into the main body of the paper as the new Section 5.
>
> ---
>
> _Figure 3 is not very informative._
>
> We have updated the paper to clarify our description of the figure.
>
> To be clear, this figure is not comparing the quality of fit and the empirical data. Rather, it is intended to show the intuition behind the invariant. Specifically, the straight lines in the log-log scale (for large enough pruning levels) illustrate that there is a power-law relation between width/depth and density for a fixed level of error.
>
> ---
>
> _In Eq. 1, it would have been useful if p had been introduced in the notation section._
>
> In the notation section, we introduce only the notation describing the models, datasets, and pruning densities. $p$ is a parameter specific to the definition of the functional form, so we introduce it in that context immediately following Eq. 1.
>
> ---
>
> _In figure 4, please add legends to the different colors in each plot._
>
> We have prioritized revising the content of the paper during the discussion period, but we will update this figure accordingly in the next revision of the paper.

---

> > ### Comment · AnonReviewer4 · 2020-11-25
> > **Improvement**
> >
> > Thanks to the authors for their response.
> >
> > I believe the paper has been improved in terms of readability, and the abstract makes the contribution more clear.
> >
> > I still have concerns around the generalizability of the method across different architectures and datasets. As it stands, the scaling law has only been shown relevant for convnets and image classification.
> >
> > If only 40 points are necessary to get a reasonable fit, it should not be impossible to add experiments covering a small MLP, LSTM Transformer or GNN on some other types of tasks (e.g. regression, sequence modelling or graph classification).

---

### Official Review · AnonReviewer5 · 2020-11-06
**Interesting Power Law Analysis of Test Error with varying Depth, Width, Dataset Size and Pruning**

**Rating:** 6
**Confidence:** 4

**Review:**

# Summary

This paper explores the possibility to fit power laws to the behaviour of  a neural network's test error under weight pruning and modifying the architecture (depth and width) and subsampling the dataset. This is interesting for practitioners when determining how to change an architecture to achieve desired performance by fitting the proposed power law to a few trained networks.

# Strong and weak points

## Pros
- The fit of the free parameters yields a very high accuracy for the architectures considered.
- The resulting parameters of the functional forms yield a direct interpretation.
- The work is self-contained and well-written

## Cons
- This is a very high bar, but let me phrase this distant goal anyway: This work does not provide an ab initio derivation of how network depth, width, dataset size and pruning are related to the test error
- There is room for considering more datasets, architectures. From my side, you don't need to do this to be accepted. But the weight of an empirical paper with structural claims grows with the number of experiments performed.

# Questions
- Do you have any failure cases for when you are not able to successfully fit the free parameters? I would imagine that e.g. different choices of hyper parameters might yield different laws.
- If the primary goal in the pruning literature is badly chosen in terms of resulting number of parameters, does it still serve as a benchmark for comparing pruning methods?

# Recommendation
This work reduces finding architecture hyperparameters by fitting parameters of a surrogate problem using a small amount of training instances. This can prove useful in practice and the surrogate problem seems to capture the main dynamics well.
This makes the work a useful tool and I recommend accepting it to ICLR.

# Minor remarks
I have some suggestions for the figures in the paper: Please make the lines in legends thicker, like the lines in Figure 6. I had to zoom in significantly to distinguish the colours in most other plots, especially for the dotted lines in Figure 4.
Instead of taking log10(X) in your figures, please use logarithmic ticks so that one can immediately grasp the values. For example, I took out my calculator to know what $10^{-1.2}$ is in Figure 1.

Maybe you can reduce the number of footnotes in the paper. Personally, I am a huge fan of in-or-out – state it in the main text if important and leave out completely if not.

---

> ### Author Response · Authors · 2020-11-22
> **Author Response**
>
> We thank the reviewer for the detailed feedback. We have responded to the reviewer’s comments in-line below.
>
> ---
>
> _This is a very high bar, but let me phrase this distant goal anyway: This work does not provide an ab initio derivation of how network depth, width, dataset size and pruning are related to the test error._
>
> Although this goal is beyond the scope of our work, we agree that delving into the origins of the phenomena we observe is an exciting direction for future research, and we are hopeful that the emergence of these scaling laws is useful for pursuing this goal.
>
> ---
>
> _There is room for considering more datasets, architectures._
>
> In Appendix E, we explore additional configurations: the VGG network family and the SynFlow pruning algorithm.
>
> In order to ensure our scaling law obtains an accurate fit across orders of magnitude (and to evaluate this fit across multiple replicates), adding a new architecture or dataset requires training an immense number of networks. We pushed our resources and our budget to the limit in order to obtain the results in the current paper.
>
> We agree that it will be exciting to explore this scaling law across additional settings and architectures, but we believe the current experiments are convincing and comprehensive for the domains we consider and we lack the resources to add more large-scale settings.
>
> ---
>
> _Do you have any failure cases for when you are not successfully able to fit the free parameters? I would imagine that, e.g., different choices of hyper parameters might yield different laws._
>
> We suspect the scaling law will not capture _one-shot_ pruning techniques (those where pruning occurs in a single step rather than iteratively). The reason is that these pruning techniques suffer from _layer collapse_ (extensively documented by Tanaka et al., 2020) where decreasing the density disproportionately prunes certain layers and prematurely disconnects the network. When this occurs, accuracy abruptly drops with lower density.
>
> Due to resource limitations, we have not explored the effect of different choices of hyperparameters on our scaling law, but we agree that this is an interesting direction for future research.
>
> ---
>
> _If the primary goal in the pruning literature is badly chosen in terms of resulting number of parameters, does it still serve as a benchmark for comparing pruning methods?_
>
> The scaling law we have found does not deal with number of parameters directly (or any other quantity for evaluating pruning for that matter), but rather it ties the generalization error to the problem constituents: the structural degrees of freedom of the architecture (width, depth), the number of samples in the dataset and the pruning level (as inducing the density of the model).
> As such, any benchmark / cost of interest (e.g. FLOPS, Latency) of these constituents $q(w,l,d)$ can in principle be reasoned about with respect to the error.
> The motivating question in its generality can be thus considered:
>
> $l,w,d = argmin_{l,w,d} q(l,w,d)$  s.t. $\epsilon_{np} \left\Vert  \left[l^\phi w^\psi d-jp'(\epsilon^\uparrow/\epsilon_{np})^{1/\gamma}\right]\cdot\left[l^\phi w^\psi d-j p'\right]^{-1} \right\Vert^\gamma=\epsilon_k$
>
> ---
>
> _Please make the lines in legends thicker._
>
> _Instead of taking log10(X) in your figures, please use logarithmic ticks._
>
> We have prioritized revising the content of the paper during the discussion period, but we will update the graphs accordingly in the next revision of the paper.

---

> > ### Comment · AnonReviewer5 · 2020-11-23
> > **Follow-Up on more datasets and architectures**
> >
> > Many thanks for your kind and detailed clarifications that answer my questions.
> >
> > Please allow me a follow-up regarding the call for more datasets and architectures that was also raised by my fellow reviewers.
> > In your answer to R2 you state: *We found that the fit variation falls quickly and that only 10s of points are needed to obtain a good fit*
> > Am I correct that you could train just a few instances per new dataset and architecture even under a constrained budget and see if the proposed power law yields useful predictions? I think that reporting on this could significantly increase the impact of this paper.
> >
> > Are Section 5 and the new abstract the only change you made to the paper? I think that it's a good idea to highlight all changes you make during the discussion using one or several colours.

---

> > > ### Author Response · Authors · 2020-11-23
> > > **Re: Follow-Up on more datasets and architectures**
> > >
> > > We can certainly train a few instances (O(10)) to validate that the scaling law is _consistent_ , i.e. that a fit on few points results in low bias $\mu$ and standard deviation $\sigma$ on those same points. If you think this would be valuable to show, we can do so in a later version of this paper for additional architectures and datasets as suggested.
> > >
> > > However, we caution that this evaluation will be less rigorous than the evaluation we conduct in Section 5, where we fit the scaling law on 10s of points but evaluate the fit on thousands of points.
> > >
> > > We highlighted all changes in the paper - thank you for the suggestion!

---

### Official Review · AnonReviewer2 · 2020-11-08
**limited applicability but good theoretical contribution**

**Rating:** 6
**Confidence:** 3

**Review:**

The paper investigates the behavior of the test error as a function of the density of the
network after pruning and identifies 3  regimes:
1) a low-density high error plateau; 2) a high-density low error plateau; 3) a power-law behavior for intermediate density.
The authors propose a functional form that captures the test error behavior in all these regimes,
along the lines of Rosenfeld et al. ICLR 2020. The approximating function contains the unpruned network's error and other 3 parameters that have to be fitted on each architecture.
The functional form had to be slightly modified with respect to Rosenfeld et al. to better describe
the density dependence near the low error transition.

Moreover, they generalize the approximation to take into account also the width and depth scaling factor,
and the dataset size as well. The generalized functional form is still relatively simple,
containing only 2 extra parameters. In fact, the authors were able to identify a roughly invariant
quantity characterizing constant error manifolds. This greatly simplifies the modeling.

Numerical experiments are performed on CIFAR10  and ImageNet datasets, using VGG and
ResNet architectures.
Two algorithms, iterative magnitude pruning and SynFlow, are used for pruning.
The proposed functional form is in good agreement with the experiments on each combination of
architecture, dataset and pruning algorithm presented.


There are a few drawbacks to the paper.
Numerical experiments involve only image classification tasks, on a small set of datasets and architectures.
Also, one of the applications motivating this kind of analysis, neural architecture search,
is in my opinion unpractical under the framework proposed, as I argue below.

On the other hand, the experiments available are quite convincing,
and the scaling form proposed is helpful in the qualitative and quantitative understanding and reasoning about the test error surface as a function of pruning and architecture.
Therefore, I suggest acceptance of the manuscript.


Major Comments
- End of Section 2:
"""
We also eliminate configurations where increasing the width or depth of the
unpruned network lowers test accuracy (e.g., 144 of the 294 CIFAR-10 ResNet configurations of l, w,
1 and n); these are typically unusual, imbalanced configurations (e.g., l = 98, w = 1/16
). Of the 12,054 possible CIFAR-10 ResNet configurations, about 8,000 are eliminated based on these sanity checks.
"""
This seems to be a very delicate point. It implies that the region of parameters for which the prediction
is a good approximation to the error is not well characterized. Minimization of the approximating function may well give configurations that lie outside those boundaries and that in practice give a much worse error.

- Appendix D. The section explains how to obtain the fit coefficient by varying only a few dimensions at a time.
Unfortunately, experimental details (e.g. number of points used) are missing.
No indication is given of the minimum number of points that one has to acquire in order to have robust fit.
The main concern here is that the fitted parameters, and the solution of the minimization problem in Section 6 as well,  may be very unstable until a very high number of points is acquired.

- Since the construction of the fitting function relies on a high number of point acquisitions to perform the fit,
it seems that the motivating question "Given a family of neural networks, which should we prune (and by how much) to obtain the network
with the smallest parameter-count such that its error does not exceed some threshold  k ?"
could be better answered by direct black-box optimization or pruning. Therefore the applicability of this
theoretical framework seems very limited.

Minor Comments:

- Fig.3: would you consider adding depth vs width? Does the invariant w*l^\alpha = v hold? I'm aware that discretization could be a problem.
- pag. 2: "To achieve sparsity levels beyond 20%" -> "To achieve density levels below 80%"
- pag. 3: "has been fit" -> "has been fitted"

---

> ### Author Response · Authors · 2020-11-22
> **Author Response**
>
> We thank the reviewer for the detailed feedback. We have responded to the reviewer’s comments in-line below.
>
> ---
>
> _Numerical experiments involve only image classification tasks, on a small set of datasets and architectures._
>
> In order to comprehensively demonstrate the validity of our scaling law across orders of magnitude and high resolution of configurations (and to evaluate this across multiple replicates), adding a new architecture or dataset requires training an immense number of networks. We pushed our resources and our budget to the limit in order to obtain the results in the current paper. We focused on image classification in particular because this is the best-studied task in the pruning literature [Blalock et al., 2020].
>
> We agree that it will be exciting to explore this scaling law across additional settings and architectures, but we believe the current experiments are convincing and comprehensive for the domains we consider and we lack the resources to add more large-scale settings.
>
> ---
>
> _[Eliminating configurations] seems to be a very delicate point. It implies that the region of parameters for which the prediction is a good approximately of the error is not very well characterized._
>
> The set of configurations we eliminate is indeed well-characterized. These are settings where the increasing width or depth causes error to increase, i.e., settings where the network has grown to the point where the error has become _data limited._ Previous work on scaling laws for unpruned networks characterize where the error becomes data limited [Rosenfeld et al. 2020, Kaplan et al 2020].
>
> ---
>
> _Minimization of the approximating function may well give configurations that lie outside those boundaries and in practice give a much worse error._
>
> Minimization is not expected to yield configurations outside of this well-characterized region because these configurations are inefficient (i.e., they are larger but are no better in terms of error).
>
> ---
>
>
> _Appendix D...explains how to obtain the fit coefficients by varying only a few dimensions at a time. Unfortunately, experimental details (e.g., number of points used) are missing...The main concern is that the fitted parameters, and the solution of the minimization problem in Section 6 as well, may be very unstable until a high number of points is used._
>
> In the former Appendix F (now Section 5 in the main body), we analyze the stability of the fit as a function of the number of points. We found that the fit variation falls quickly and that only 10s of points are needed to obtain a good fit.
>
> We have also updated Appendix D to specify the number of points used.
>
> ---
>
> _Since the construction of the fitting function relies on a high number of point acquisitions to perform the fit, it seems that the motivating question...could be better answered by direct black-box optimization or pruning. Therefore the applicability of this theoretical framework seems very limited._
>
> We distinguish between two different tasks:
> 1. Constructing the scaling law (i.e., finding its form)
> 2. Fitting the scaling law (i.e., fitting the five parameters once we already know its form)
>
> Task 1 required a large amount of data to find the form of the scaling law and rigorously verify its accuracy.
>
> As we show in the new Section 5, Task 2 requires pruning only a small number of networks once we know the form of the scaling law.
>
> ---
>
> _One of the applications motivating this kind of analysis, neural architecture search, is in my opinion unpractical under the framework proposed._
>
> While we did not envision or argue for immediately applying the current work to NAS, we believe that our contributions could provide a useful stepping stone toward the broader effort of exploiting the structure inherent to pruning and its relationship to architecture in the context of NAS. We hope our clarifications above (about the configurations eliminated and the number of points needed to fit the scaling law) substantiate this belief.
>
> ---
>
> _Fig 3: would you consider adding depth vs width? Does the invariant_ $w l^\alpha = v$ _hold?_
>
> It depends on the density of the model. When the model is not heavily pruned (or is not pruned at all), the invariant $w l^\alpha = v$ does not hold. However, for heavily enough pruned models,  $w l^\alpha = v$ is a direct result of $w^\psi l^\phi d = v$ (by setting $\alpha = (\phi/\psi)$:  $w l^{(\phi/\psi)} = (v d)^\psi = v''$ for constant $d$).
>
> We did not have time to add the graph of depth vs. width during the author response period, but we will do so for the next revision of the paper.
>
> ---
>
> _Corrections and grammar._
>
> Thank you for pointing these out. We have updated the paper accordingly.

---

### Decision · Program_Chairs · 2021-01-07
**Final Decision**

**Decision:**

Reject

**Comment:**

The paper provides a functional approximation of the error of ResNets and VGGs pruned with IMP and SynFlow on CIFAR-10 and ImageNet, showing that it is predictable in terms of an invariant tying width, depth, and pruning level.  In particular, it formulates the test error as a function of the density of the network after pruning and identifies a low-density high error plateau, a high-density low error plateau, and a power-law behavior for intermediate density. It further demonstrates that networks of different sparsities are freely interchangeable. The paper provides an interesting insight on the power law structure of the error as networks are pruned, however the results are very limited to specific types of networks (ResNets and VGGs), pruning methods (IMP and SynFlow) and datasets (CIFAR-10, ImageNet). Hence, it's not clear if the proposed functional approximation generalizes to other network families, pruning methods, and datasets. I understand that adding a new architecture or dataset is expensive, but fitting the proposed scaling law (the five parameters) requires pruning only a small number of networks, as mentioned by the authors. Comparing the calculated error and the actual error of the pruned network for different architectures and datasets can help verify the findings in the paper, and significantly widens its scope.